# One health at the last mile: Multi-scale predictors of *Schistosoma japonicum* infection in southwest China across two decades of control

William W. Zou[1], Elise N. Grover[1], Liu Yang[2], Elizabeth J. Carlton[1]*

**1** Department of Environmental and Occupational Health, Colorado School of Public Health, University of Colorado Anschutz, Aurora, Colorado, United States of America, **2** Sichuan Center for Disease Control and Prevention, Chengdu, Sichuan, China

\* elizabeth.carlton@cuanschutz.edu

## Abstract

In China, schistosomiasis is targeted for elimination. As the country approaches elimination, it is critical to evaluate how the dynamics of transmission are changing in remaining pockets of disease. We have been studying areas of schistosomiasis reemergence and persistence in Sichuan, China since 2007. This study used gradient boosting machines to identify key predictors of infection across two periods, 2007–2010, a period when schistosomiasis had reemerged, and 2016–2019, a period when schistosomiasis was approaching elimination. We also evaluated how key predictors of infection have shifted over time and whether combinations of predictors amplified risk. We considered predictors describing agriculture, domestic animals, socio-economic status, water and sanitation infrastructure, and demographics at individual, household and village-level scales. Our re-emergence and elimination models demonstrated strong predictive performances (AUC-PR = 0.92 and AUC = 0.85, respectively). In both periods, a person's age and village level agricultural practices including the average area of dry crops, rice planted, and night soil use, were among the most influential factors. Village-level factors dominated in 2007–2010, while household and individual predictors increased in predictive importance in 2016–2019. Between 2007–2010 and 2016–2019, there were increases in the importance of household agricultural practices such as the area of dry crops and rice cultivated, and household cat and dog ownership, while the importance of factors describing water and sanitation infrastructure decreased. In the elimination period, our models found the combination of high village dry crop cultivation and lack of improved sanitation amplified infection probability. Our findings suggest adding precision interventions targeting high-risk households on top of existing community-wide measures may accelerate schistosomiasis elimination. Practitioners should consider adding agricultural, sanitation and animal infection data to end-game surveillance programs, while

**Data availability statement:** The data used in this study contains protected health information (PHI) and cannot be shared publicly due to participant privacy and confidentiality restrictions. Data access for researchers who meet the criteria for access to confidential data may be made available upon reasonable request, subject to institutional and ethical approvals. Requests for data access should be directed to the Colorado Multiple Institutional Review Board (COMIRB) comirb@ucdenver.edu.

**Funding:** WZ, EG, LY and EJC were supported by the National Institute of Allergy and Infectious Diseases of the National Institutes of Health under Award Number R01AI134673 to EJC. LY and EJC were also supported by R21AI115288 to EJC, and R01AI068854 which supported data collection from 2007 to 2010. The content is solely the responsibility of the authors and does not necessarily represent the official views of the National Institutes of Health. The funders played no role in the study design, data collection and analysis, decision to publish, or preparation of the manuscript.

**Competing interests:** The authors have declared that no competing interests exist.

researchers evaluate the consistency of these findings in other low-endemic settings and explore causal pathways to inform adaptive, locally tailored strategies.

## Author summary

Schistosomiasis is a parasitic disease that has been a target of disease control efforts globally, with China aiming to eliminate the disease as a public health problem. In China, disease control efforts have been successful in reducing the spread and prevalence of the disease, though there are remaining pockets of low levels of transmission. Our study compared the most important predictors of schistosomiasis infection risk between two periods, 2007–2010, a period when schistosomiasis had reemerged, and 2016–2019, a period when schistosomiasis was approaching elimination. We found village-level factors were the most important predictors of infection in the earlier and later periods, while household and individual-level increased in importance in the later period. For example, the area rice and other crops cultivated were positively associated with infection. The importance of potential animal hosts such as ownership of cats and dogs also increased over time. We also found that the age of peak disease risk shifted from 40-60 to >80 years of age over our 13-year study period. Our results indicate that the factors behind disease may be changing, potentially due to the selective pressures of decades of disease control and largescale socioeconomic changes such as urbanization.

## Introduction

As the World Health Organization (WHO) pushes for global schistosomiasis elimination by 2030, there is an urgency to understand the epidemiological dynamics that characterize the final stages of disease control [1]. Schistosomiasis is caused by multiple human-infecting *Schistosoma* species, with *Schistosomiasis mansoni* found in Africa and the Americas, *S. haematobium* in Africa, and *S. japonicum* in Asia [2]. Together, these parasites impose a substantial global disease burden, driven by persistent transmission in settings shaped by poverty, surface water contact, and limited sanitation infrastructure. WHO's goal for schistosomiasis is "elimination as a public health problem," operationalized as reducing the prevalence of heavy-intensity infections to <1% [1]. The edge of elimination for schistosomiasis remains open and under-explored: do the drivers of transmission shift? Do interventions need to be adapted as prevalence declines? Identifying who remains at risk and which factors sustain transmission is important for effective control efforts in the face of changing ecological, economic, behavioral, and demographic conditions.

China offers a strong case study for investigating these questions. Intestinal schistosomiasis caused by *S. japonicum* is a zoonotic disease system, with humans and a wide range of non-human mammals capable of serving as definitive hosts, potentially contributing to environmental contamination and human risk [3]. In the *S.*

*japonicum* life cycle, eggs excreted in mammalian feces hatch in freshwater and infect amphibious *Oncomelania* snails, which eventually release cercariae and perpetuate mammalian infection risk upon contact with contaminated water [4]. Once hyperendemic for intestinal schistosomiasis with a seroprevalence of 34.8% in 1982, China has achieved largescale reductions through a national control program involving mass drug administration, snail control, and targeted treatment of bovine reservoirs [5].By 2020, seroprevalence had dropped to 2.4%, and the number of endemic counties had declined substantially [6]. Despite this progress, schistosomiasis remains endemic in 113 counties as of 2023 [7], including the rural and mountainous regions of Sichuan province, where seroprevalence remains at 1.12% [8]. While humans and bovines have long been the primary hosts targeted for control, emerging evidence suggests that other animals, such as dogs, pigs and rodents, may also contribute to ongoing transmission [9–10]. Simultaneously, broader socio-economic shifts, including an aging rural population and land-use change, may be reshaping patterns of exposure in ways that legacy interventions do not fully address [11–12]. Understanding how these factors impact infections, whether their importance has shifted over time, and the extent to which different exposures interact to amplify risk is critical for refining China's control strategy and for informing global efforts in similar late-stage landscapes.

To capture these evolving dynamics, we used infection, household and demographic data from 2007 to 2019 in rural Sichuan, China and analyzed it using Gradient Boosting Machines (GBMs) to compare predictors of infection during two critical time periods: 2007–2010, when schistosomiasis reemerged in parts of Sichuan [13], and 2016–2019, when the country approached its elimination targets [14]. These methods are well-suited for evaluating a suite of potential risk factors, modeling complex, non-linear relationships and interactions between predictors, and build on prior work by Grover et al. and Buchwald et al. [15–17]. Machine learning approaches offer new opportunities to uncover key predictors of infection alone and in combination, and explore how the relative importance of environmental, agricultural, water, sanitation and hygiene (WASH), and socio-economic variables has changed in response to decades of control [18]. By evaluating shifts in predictor importance, we aim to provide insight into the changing dynamics of transmission and identify optimal conditions for human infections to occur. While our analysis is specific to *S. japonicum* in Sichuan, China, we also aim to provide a model for identifying key predictors of infection in other contexts approaching elimination that can be used to inform surveillance strategies approaching and the design and evaluation of targeted interventions.

## Materials and methods

### Ethics statement

This study was approved by the Sichuan Institutional Review Board (China), the University of California, Berkeley (USA), Committee for the Protection of Human Subjects, and the Colorado Multiple Institutional Review Board (USA). All participants provided written, informed consent. Children provided assent and their parents or guardians provided written informed permission for their children to participate in the study.

### Study region and design

Data were collected in 2007, 2010, 2016, and 2019 in Sichuan, China, as part of an ongoing study of schistosomiasis reemergence and persistence in areas targeted for elimination. The study has focused on areas where the disease was suspected to be present despite aggressive local control measures, adding new areas of suspected transmission over time. It is not intended to provide a representative sample of Sichuan province. Details of village selection and survey methods have been described in detail elsewhere [16] and in the appendix.

### Human infection data

For each of the study years, infection status was determined by examining up to three stool samples over three consecutive days using the miracidium hatching test, examining 30 grams of stool per sample [13]. In 2007 and 2010 one stool

sample was also tested using the Kato-Katz thick smear procedure examining three slides per sample [13]. Infection surveys were conducted in November and December of 2007, 2010 and 2019, and in June and July of 2016 [13]. People were classified as infected if any of the tests were positive. Individuals who tested positive were notified and referred to the local schistosomiasis control station for treatment.

**Infection risk factors**

Data on potential risk factors were collected from household and demographic surveys. During the summers of 2007, 2010, 2016 and 2019, the head of each participating household completed a survey with close-ended questions regarding socio-economic status, domestic and farm animal ownership, sanitation and water access, and agricultural practices. Demographic surveys were conducted as part of the census and administered to the participant directly, or to the head of household. In 2010, demographics surveys were only administered to new participants. All surveys were conducted in the local dialect by trained staff from the Sichuan Center for Disease Control and Prevention and the county's Schistosomiasis Control Stations.

We selected candidate predictors of human infection based on six categories that we hypothesized could contribute to human *S. japonicum* infection risk: 1) agricultural practices, 2) animal reservoirs, 3) individual demographics characteristics, 4) occupational risk factors, 5) socioeconomic status (SES), and 6) WASH indicators (Table 1). Additionally, we included a county predictor to account for differences in schistosomiasis control program administration and other differences between counties not captured in the aforementioned risk factors.

In the study region, common agricultural practices include growing a range of crops throughout the year (e.g., rice, corn, vegetables, wheat, rapeseed) and using night soil (a mix of human and animal waste extracted from stool pits), which is applied as an agricultural fertilizer, in addition to chemical fertilizers. We included estimates of total annual crop area, and the annual volume of night soil applied to crops as predictors of human infection. Because rice farming involves flooding fields in ways that can lead to snail habitat formation and distinct water contact patterns, we also categorized crop area and night soil use into two groups: "rice crops" and "dry crops" (i.e., all other crops).

We generated a nine-point composite asset score as a proxy for SES, following Grover et al. [16]. Household assets were derived from the household survey, wherein the head of household was asked whether they owned any of `the following eight assets: tractors, televisions, air conditioners, refrigerators, computers, cars or trucks, motorcycles, and washing machines. The asset score also included an indicator of whether the home was constructed from either brick or concrete (as opposed to adobe). The composite score was calculated by summing up all reported assets for a given household, yielding a score between zero and nine.

We developed village-level variables that summarized agricultural, animal reservoir, SES and WASH related risks to provide broader context on community-wide exposure and risk factors and capture environmental and socio-economic influences that may impact schistosomiasis infection risk beyond the individual or household scale. Village-level variables were constructed from household survey data collected from all other households located in the same village, excluding data from the household itself to avoid interdependence between household-level and village-level variables. We aggregated continuous household variables to village-mean values and binary household variables to a village-proportion value.

We established criteria for including, removing, or collapsing variables. Only predictors that were available in all study years were included in this analysis. We decided *a priori* to exclude or transform variables with less than 10% or over 90% of observations in a single category. For example, for variables like occupation where the original formulation included several rare (<10%) categories, we opted for a binary formulation of the variable (farmer vs. non-farmer). We also removed highly collinear variables. For example, "House material" was excluded from this analysis because it was already a component of our SES asset score variable. We evaluated multicollinearity using Pearson correlation coefficients and planned to select representative predictors from clusters of predictors with values >0.7 or <-0.7 but no variables had correlations above the threshold [19; S2 Fig]. The final dataset included 25 predictors: one county-level, four individual-level, ten household-level, and ten village-level variables (Table 1).

**Table 1. Description of candidate predictors for *S. japonicum* infection by year, including variables that describe agricultural practices, potential animal reservoirs, individual characteristics, occupation, socio-economic status, and access to water, sanitation and hygiene (WASH) infrastructure.**

| | Predictor Category | 2007 Total Tested | 2007 N Positive | 2007 % Positive | 2010 Total Tested | 2010 N Positive | 2010 % Positive | 2016 Total Tested | 2016 N Positive | 2016 % Positive | 2019 Total Tested | 2019 N Positive | 2019 % Positive |
|---|---|---|---|---|---|---|---|---|---|---|---|---|---|
| **Total** | | 1974 | 167 | 8.46% | 1126 | 98 | 8.70% | 612 | 49 | 8.00% | 689 | 7 | 1.02% |
| Rice Area* (H) (Mus†) | Agricultural Practices | | | | | | | | | | | | |
| 0.00-0.60 | | 557 | 32 | 5.75% | 400 | 19 | 4.75% | 228 | 13 | 5.70% | 336 | 3 | 0.89% |
| 0.61-1.60 | | 767 | 67 | 8.74% | 354 | 43 | 12.15% | 175 | 19 | 10.86% | 159 | 0 | 0.00% |
| 1.61-25.00 | | 650 | 68 | 10.46% | 372 | 36 | 9.68% | 208 | 17 | 8.17% | 192 | 4 | 2.08% |
| Rice Area* (V) (mean Mus) | Agricultural Practices | | | | | | | | | | | | |
| 0.00-0.84 | | 451 | 10 | 2.22% | 436 | 22 | 5.05% | 234 | 25 | 10.68% | 347 | 3 | 0.87% |
| 0.85-1.44 | | 796 | 73 | 9.17% | 284 | 38 | 13.38% | 195 | 9 | 4.62% | 192 | 2 | 1.04% |
| 1.45-3.50 | | 727 | 84 | 11.55% | 406 | 38 | 9.36% | 183 | 15 | 8.20% | 150 | 2 | 1.33% |
| Dry Crop Area* (H) (Mus) | Agricultural Practices | | | | | | | | | | | | |
| 0.00-3.00 | | 667 | 44 | 6.60% | 331 | 19 | 6.11% | 185 | 21 | 11.35% | 440 | 3 | 9.68% |
| 3.01-5.30 | | 679 | 55 | 8.10% | 335 | 29 | 8.66% | 164 | 7 | 4.27% | 168 | 2 | 1.19% |
| 5.31-65.90 | | 628 | 68 | 10.83% | 480 | 50 | 10.42% | 263 | 21 | 7.99% | 81 | 2 | 2.47% |
| Dry Crop Area* (V) (mean Mus) | Agricultural Practices | | | | | | | | | | | | |
| 0.05-3.10 | | 665 | 44 | 6.62% | 194 | 6 | 3.09% | 137 | 22 | 16.06% | 474 | 4 | 0.84% |
| 3.11-5.03 | | 580 | 45 | 7.76% | 490 | 55 | 11.2=22% | 195 | 9 | 4.62% | 199 | 2 | 1.01% |
| 5.04-14.14 | | 729 | 78 | 10.70% | 442 | 37 | 8.37% | 280 | 18 | 6.43% | 16 | 1 | 5.26% |
| Night Soil Rice* (H) (buckets) | Agricultural Practices | | | | | | | | | | | | |
| 0-6 | | 1378 | 105 | 7.62% | 965 | 85 | 8.81% | 535 | 38 | 7.10% | 643 | 7 | 1.09% |
| 7-30 | | 295 | 20 | 6.78% | 114 | 8 | 7.02% | 49 | 6 | 12.25% | 32 | 0 | 0.00% |
| 31-400.00 | | 301 | 42 | 13.95% | 47 | 5 | 10.64% | 28 | 5 | 17.86% | 14 | 0 | 0.00% |
| Night Soil Rice* (V) (mean buckets) | Agricultural Practices | | | | | | | | | | | | |
| 0.00-2.00 | | 167 | 5 | 2.99% | 666 | 64 | 9.61% | 241 | 12 | 4.98% | 400 | 6 | 1.50% |
| 2.01-9.00 | | 704 | 22 | 3.13% | 235 | 8 | 3.404% | 234 | 15 | 6.41% | 289 | 1 | 0.35% |
| 9.01-67.00 | | 1103 | 140 | 12.69% | 225 | 26 | 11.56% | 137 | 22 | 16.06% | NA | NA | NA |
| Night Soil Dry Crops* (H) (buckets) | Agricultural Practices | | | | | | | | | | | | |
| 0-25 | | 1131 | 83 | 7.34% | 826 | 81 | 9.81% | 473 | 39 | 8.25% | 522 | 6 | 1.15% |
| 26-1500 | | 843 | 84 | 9.96% | 300 | 17 | 5.67% | 139 | 10 | 7.19% | 167 | 1 | 0.60% |
| Night Soil Dry Crops* (V) (mean buckets) | Agricultural Practices | | | | | | | | | | | | |
| 0.00-16.00 | | 351 | 32 | 9.12% | 541 | 55 | 10.17% | 245 | 11 | 4.49% | 338 | 3 | 0.89% |
| 16.01-35.00 | | 693 | 31 | 4.47% | 420 | 39 | 9.29% | 160 | 15 | 9.38% | 189 | 1 | 0.53% |
| 35.01-243.00 | | 930 | 104 | 11.18% | 165 | 4 | 2.42% | 207 | 23 | 11.11% | 162 | 3 | 1.85% |

*(Continued)*

**Table 1.** (Continued)

| | Predictor Category | 2007 | | | 2010 | | | 2016 | | | 2019 | | |
|---|---|---|---|---|---|---|---|---|---|---|---|---|---|
| | | Total Tested | N Positive | % Positive | Total Tested | N Positive | % Positive | Total Tested | N Positive | % Positive | Total Tested | N Positive | % Positive |
| Bovines (H) (number of cows and buffalo) | Animal Reservoirs | | | | | | | | | | | | |
| 0 | | 1760 | 146 | 8.30% | 1009 | 87 | 8.62% | 565 | 47 | 8.32% | 670 | 6 | 0.90% |
| 1-16 | | 214 | 21 | 9.81% | 117 | 11 | 9.40% | 47 | 2 | 4.26% | 19 | 1 | 5.26% |
| Bovines* (V) (mean number of cows and buffalo) | Animal Reservoirs | | | | | | | | | | | | |
| 0.00-0.20 | | 353 | 7 | 1.98% | 469 | 33 | 7.04% | 222 | 23 | 10.36% | 479 | 5 | 1.04% |
| 0.21-0.52 | | 643 | 49 | 7.62% | 265 | 18 | 6.79% | 343 | 25 | 7.29% | 180 | 2 | 1.11% |
| 0.53-1.75 | | 978 | 111 | 11.35% | 392 | 47 | 11.99% | 47 | 1 | 2.13% | 30 | 0 | 0.00% |
| Cats (H) (ownership: y/n) | Animal Reservoirs | | | | | | | | | | | | |
| No | | 939 | 75 | 7.89% | 592 | 47 | 7.94% | 285 | 16 | 5.61% | 486 | 6 | 1.24% |
| Yes | | 1025 | 92 | 8.97% | 534 | 51 | 9.55% | 322 | 31 | 9.63% | 203 | 1 | 0.49% |
| Cats* (V) (prevalence) | Animal Reservoirs | | | | | | | | | | | | |
| 0.00-36.67% | | 366 | 22 | 6.01% | 475 | 46 | 9.68% | 111 | 4 | 3.60% | 532 | 6 | 1.13% |
| 36.68-51.06% | | 741 | 58 | 7.83% | 342 | 27 | 7.90% | 292 | 28 | 9.59% | 91 | 0 | 0.00% |
| 51.07-87.50% | | 867 | 87 | 10.4% | 309 | 25 | 8.09% | 292 | 17 | 9.59% | 66 | 1 | 1.52% |
| Dogs (H) (ownership: y/n) | Animal Reservoirs | | | | | | | | | | | | |
| No | | 437 | 30 | 6.87% | 265 | 20 | 7.55% | 171 | 12 | 7.02% | 276 | 1 | 0.36% |
| Yes | | 1527 | 137 | 8.97% | 861 | 78 | 9.06% | 437 | 36 | 8.24% | 413 | 6 | 1.45% |
| Dogs* (V) (prevalence) | Animal Reservoirs | | | | | | | | | | | | |
| 0.00-67.33% | | 423 | 24 | 5.67% | 378 | 34 | 9.00% | 299 | 29 | 9.70% | 436 | 5 | 1.15% |
| 67.34-79.40% | | 729 | 49 | 6.72% | 331 | 14 | 4.23% | 199 | 18 | 9.05% | 149 | 1 | 0.67% |
| 79.41-100.00% | | 822 | 94 | 11.44% | 417 | 50 | 11..99% | 114 | 2 | 1.75% | 104 | 1 | 0.96% |
| Sex (I) | Demographics | | | | | | | | | | | | |
| Female | | 969 | 87 | 8.98% | 512 | 55 | 10.74% | 296 | 29 | 9.80% | 337 | 3 | 0.89% |
| Male | | 1005 | 80 | 7.96% | 582 | 41 | 7.05% | 315 | 20 | 6.35% | 344 | 4 | 1.16% |
| Age* (I) | Demographics | | | | | | | | | | | | |
| 6.00-42.67 | | 889 | 65 | 7.31% | 370 | 24 | 6.49% | 150 | 6 | 4.00% | 50 | 0 | 0.00% |
| 42.68-56.17 | | 694 | 63 | 9.08% | 387 | 40 | 10.34% | 169 | 16 | 9.47% | 204 | 4 | 1.96% |
| 56.18-95.83 | | 390 | 39 | 10.00% | 336 | 32 | 9.52% | 292 | 27 | 9.25% | 427 | 3 | 0.70% |
| Occupation (I) | Occupational Risk Factor | | | | | | | | | | | | |
| Farmer and Fisher | | 1765 | 158 | 8.95% | 970 | 90 | 9.28% | 4990 | 44 | 8.98% | 632 | 7 | 1.11% |
| Other Occupations | | 207 | 9 | 4.35% | 115 | 3 | 2.61% | 108 | 5 | 4.63% | 44 | 0 | 0.00% |

*(Continued)*

| | Predictor Category | 2007 | | | 2010 | | | 2016 | | | 2019 | | |
|---|---|---|---|---|---|---|---|---|---|---|---|---|---|
| | | Total Tested | N Positive | % Positive | Total Tested | N Positive | % Positive | Total Tested | N Positive | % Positive | Total Tested | N Positive | % Positive |
| Education (I) | SES | | | | | | | | | | | | |
| None | | 303 | 31 | 10.23% | 186 | 21 | 11.29% | 137 | 16 | 11.68% | 181 | 2 | 1.11% |
| Elementary School | | 1020 | 92 | 9.02% | 573 | 49 | 8.55% | 297 | 24 | 8.08% | 350 | 4 | 1.14% |
| Middle School or Higher | | 637 | 43 | 6.75% | 321 | 22 | 6.85% | 160 | 8 | 5.00% | 145 | 1 | 0.69% |
| Assets (H) | SES | | | | | | | | | | | | |
| 0-3 | | 1314 | 129 | 9.82% | 370 | 50 | 13.51% | 122 | 11 | 9.02% | 97 | 1 | 1.03% |
| 4-5 | | 630 | 37 | 5.87% | 627 | 43 | 6.86% | 299 | 25 | 8.36% | 416 | 5 | 1.20% |
| 6-9 | | 30 | 1 | 3.33% | 129 | 5 | 3.88% | 191 | 13 | 6.81% | 176 | 1 | 0.57% |
| Assets* (V) (mean assets) | SES | | | | | | | | | | | | |
| 1.81-3.29 | | 756 | 58 | 7.67% | 436 | 39 | 8.94% | 170 | 4 | 2.35% | 160 | 0 | 0.00% |
| 3.30-3.87 | | 772 | 59 | 7.64% | 371 | 28 | 10.20% | 223 | 23 | 10.30% | 128 | 1 | 0.78% |
| 3.88-5.33 | | 446 | 50 | 11.20% | 319 | 21 | 6.58% | 219 | 22 | 10.00% | 401 | 6 | 1.50% |
| Well Water (H) (use: y/n) | WASH | | | | | | | | | | | | |
| No | | 549 | 49 | 8.93% | 411 | 35 | 8.52% | 293 | 10 | 3.41% | 286 | 3 | 1.05% |
| Yes | | 1425 | 118 | 8.28% | 707 | 62 | 8.77% | 317 | 39 | 12.30% | 403 | 4 | 0.99% |
| Well Water* (V) (prevalence) | WASH | | | | | | | | | | | | |
| 0.00-45.83% | | 465 | 35 | 7.53% | 426 | 44 | 10.33% | 319 | 12 | 3.76% | 265 | 1 | 0.38% |
| 45.84-90.00% | | 793 | 83 | 10.47% | 367 | 11 | 3.00% | 156 | 15 | 9.62% | 177 | 4 | 2.26% |
| 90.01-100.00% | | 716 | 49 | 6.84% | 333 | 43 | 12.91% | 137 | 22 | 16.06% | 247 | 2 | 0.81% |
| Improved Sanitation‡ (H) (ownership: y/n) | WASH | | | | | | | | | | | | |
| No | | 1593 | 113 | 7.09% | 776 | 70 | 9.02% | 302 | 32 | 10.60% | 307 | 3 | 0.98% |
| Yes | | 381 | 53 | 14.17% | 350 | 28 | 8.00% | 310 | 17 | 5.48% | 382 | 4 | 1.05% |
| Improved Sanitation* (V) (prevalence) | WASH | | | | | | | | | | | | |
| 0.00-12.00% | | 1047 | 59 | 5.64% | 296 | 9 | 3.04% | 47 | 1 | 2.13% | 79 | 2 | 2.53% |
| 12.01-47.06% | | 660 | 68 | 10.30% | 511 | 58 | 11.35% | 207 | 35 | 16.91% | 100 | 2 | 2.00% |
| 47.07-100.00% | | 267 | 40 | 14.98% | 319 | 31 | 9.72% | 358 | 13 | 3.63% | 510 | 2 | 0.59% |
| County | County Differences | | | | | | | | | | | | |
| 1 | | 1129 | 105 | 9.30% | 616 | 74 | 12.01% | 322 | 41 | 12.73% | 358 | 4 | 1.12% |
| 2 | | 845 | 62 | 7.34% | 510 | 24 | 4.81% | 290 | 8 | 2.76% | 331 | 3 | 0.91% |

*Continuous variables were categorized by tertiles for the purpose of this table and modeled as continuous variables in the statistical models.

†Mus corresponds to 1/15 of a hectare, or about 666.67 m$^2$

‡Improved sanitation here means the presence of an anaerobic biogas digester or a three-compartment toilet instead of a simple toilet or no toilet at all.

## Statistical analysis

We used GBMs to identify key predictors of schistosomiasis infection risk, evaluate their relationships with infection outcomes, and assess interactions between predictors. GBMs combine multiple regression trees into an ensemble, enabling them to fit complex, non-linear relationships, which is particularly useful for capturing the non-linear dynamics of disease transmission. Compared to traditional regression models, GBMs can provide stronger predictive performance when modeling biological processes like schistosomiasis transmission, which often involve complex interdependencies [15]. Models were constructed in R using the "gbm" and "dismo" packages [20,21].

We bifurcated the data to evaluate potential changes in infection risk factors between 2007–2010 and 2016–2019. The first period (2007–2010) covers the "reemergence" period, the period shortly after *S. japonicum* reemergence was recognized in the region, and a period when infections were relatively higher (human infection prevalence 8.63% in 2007 and 2010). The second period (2016–2019) covers the "elimination" period, a period when infections were increasingly rare (human infection prevalence 4.51% in 2016 and 2019) and *S. japonicum* elimination was being aggressively pursued.

We partitioned the two datasets into spatially balanced (similar proportion of observations across villages) and temporally balanced (similar proportion of observations across the years) training (70%), evaluation (20%) and test (10%) sets for the analysis to prevent target leakage and improve the generalizability of the results. This was performed using the "BalancedSampling" package in R [22]. We employed 5-fold cross-validation on our training sets so that the model's performance was assessed across different subsets of the data to reduce the risk of overfitting and improve generalizability.

To address class imbalance in our outcome (8.58% infection in the reemergence period; 4.51% infection in the elimination period), we over-sampled the minority class using Random Walk Oversampling (RWO), implemented via the "imbalance" package [23], which generated synthetic infected observations based on the variance and mean of the infected observations.

The datasets also contained missing values across several predictors (3.27% of data were missing due to ambiguous or incomplete responses from survey participants). We addressed missing data in the training datasets using the "randomForest" package [24] to impute missing values based on the median value for continuous variables and the mode value for binary variables.

Optimal model hyperparameters including learning rate (0.01-0.3), tree complexity (1–5), and number of trees (900–1000) were selected from a range of values using a grid search which iterated over one hundred combinations of hyperparameter values and allowed us to identify the strongest combination for each model. We assessed model performance using Area Under the Precision-Recall (AUC-PR) curves, sensitivity, specificity, accuracy, and kappa values. We evaluated our models based on AUC-PR it is robust to imbalanced datasets [25]. To account for model uncertainty, we refit the full modeling pipeline across 20 bootstrapped iterations using different random seeds and used the set of performance metrics to generate confidence intervals. In addition, because most imputation methods include a stochastic component changing the random seed can yield slightly different imputed values, especially when missingness is non-trivial, so refitting the full pipeline across 20 bootstrap iterations with different seeds tests whether results are robust to imputation-induced variability.

Our primary analytical goal was to rank predictor importance by relative importance, defined as the proportion of the total reduction in squared error that each variable contributes to the models. As secondary goals, we aimed to describe (i) the shape of non-linear marginal effects and (ii) the strength of two-way interactions in relation to *S. japonicum* infection risk.

For the six most influential predictors from each time period, we graphed these functions as Partial Dependence Plots (PDPs) using the pdp package [26]. Uncertainty around each curve was quantified with 95% confidence bands generated from 1,000 bootstrap replicates of the training data and reflect the range of predicted infection probabilities at each predictor value. These plots display the average predicted outcome (in this case, probability of infection) as a function of a single predictor, marginalizing the joint distribution of all other variables in the study. The resulting curves reflect nonlinear relationships between our predictors and the marginal probability of infection, as estimated by our BRT models.

We examined pairwise interactions between predictors because transmission dynamics are shaped by complex interdependencies among environmental, socio-economic, and biological factors. Pairwise interactions were quantified using Elith et al.'s customized "Boosted Regression Tree (BRT)" package [15]. All possible pairwise interactions were made available to the model, but they were not all forced into the final fit. Instead, only those splits that lower out-of-bag deviance while applying shrinkage, a small tree depth, and stochastic subsampling were included in the final BRT model. Regularization settings were included to limit model complexity and overfitting and improve generalizability. The most important interactions were identified using interaction size, a value that functions similarly to relative importance in that it quantifies the strength of pairwise interactions based on the additional variance explained by allowing the two variables to interact, beyond their additive effects alone. The interaction sizes are unitless and do not follow a fixed scale, allowing them to identify relatively strong interactions within a single model. However, they are not directly comparable across models or to relative importance scores, as they reflect localized interaction effects rather than global predictive contributions. We selected an interaction size of >5 as a threshold for reporting. We created three-dimensional partial plots using the "BRT" package for the top three most important interactions to examine the direction of these relationships.

## Results

Our analysis included 3,033 observations from people aged 6–96 years old across 2 rural counties and 51 villages in Sichuan province, collected from 2007 to 2019. In the study villages, populations ranged from 7-70 households and 12–156 residents. There was a marked decline in infections from 2007 to 2019, even with our sampling strategy focused on areas thought to have ongoing transmission: in 2007, 8.46% of the 1,974 people tested were infected, and by 2019, 1.02% of the 689 people tested were infected (Table 1).

Our models assessing predictors of schistosomiasis infection during the re-emergence and elimination periods demonstrated strong predictive performance, with mean AUC-PR values of 0.92 and 0.85, respectively. Both models also had high mean specificity (98% and 96%) and accuracy (0.97 and 0.95) (Table 2). Mean model sensitivity and Kappa values were higher for the re-emergence period (Sensitivity = 0.90, Kappa = 0.95) than the elimination period (Sensitivity = 0.79, Kappa = 0.88).

In the reemergence period (2007–2010), the three most important variables (ranked 1–25 with 1 being the most important) were all village-level variables: dry crop area (V), rice area (V) and night soil rice (V) (Fig 1). Age (I), improved sanitation (V), dry crop area (H) and bovine ownership (V) were also strong predictors of infection risk. Of the top seven predictors, five were village-level, one was at the household-level, and one was at the individual-level. The least important were all household-level predictors (improved sanitation, cat ownership, well water usage, and dog ownership).

During the elimination period (2016–2019), dry crop area (V), night soil rice (V), age (I) and dry crop area (H) remained strong predictors of infection (Fig 2). Meanwhile, there were large increases (change of ≥7 ranks) in the ranked importance of cat ownership (H), education (I), and dog ownership (H), assets (V), and improved sanitation (H), all of which were previously ranked low. Of the seven most important predictors, three were village-level, two were household-level, and two were individual-level predictors. The least influential predictors (ranked ≥20) were well water usage (H & V), occupation (I), county, bovine ownership (H), sex (I), night soil dry crops (H), cat ownership (V).

**Table 2. Predictive performance of the reemergence (2007-2010) and elimination models (2016-2019) with 95% confidence intervals across 20 bootstrapped iterations.**

| Year | AUC-PR | Specificity | Sensitivity | Accuracy | Kappa |
|---|---|---|---|---|---|
| 2007-2010 | 0.92 (0.90-0.93) | 0.98 (0.97-1.00) | 0.90 (0.88-0.91) | 0.97 (0.95-1.00) | 0.95 (0.80-0.99) |
| 2016-2019 | 0.85 (0.80-0.89) | 0.96 (0.92-1.00) | 0.79 (0.70-0.85) | 0.95 (0.90-1.00) | 0.88 (0.75-0.96) |

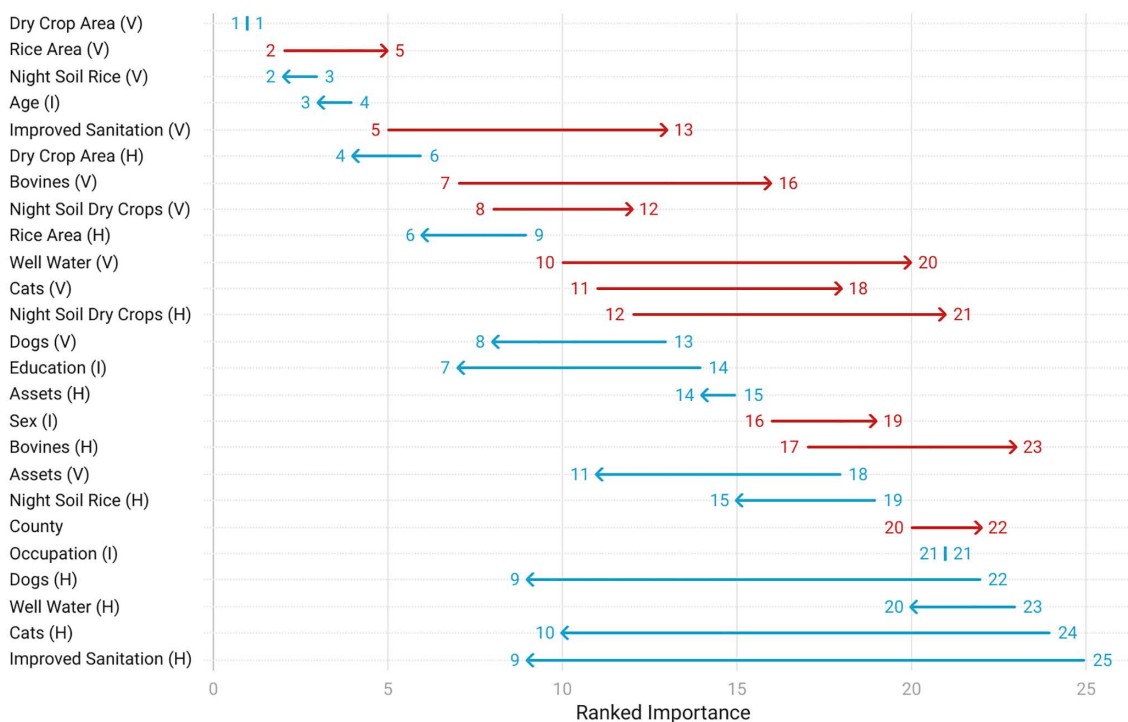

**Fig 1. Change in the ranked importance of predictors from 2007-2010 (reemergence period) to 2016-2019 (elimination period).** Red lines indicate a decrease in ranked importance from the reemergence period to the elimination period. Blue lines indicate an increase.

We saw evidence that the relationships between predictors and infection risk displayed a mixture of linear and non-linear associations. In 2007–2010, the marginal change in the probability of infection increased when dry crop area (V) exceeded 8 Mus, or when rice crop area (V) exceeded 2.5 Mus (Fig 2). We found evidence of a positive monotonic relationship between infection risk and night soil use on rice (V), rising from an infection probability of 0.01% when 0 buckets of night soil were used on rice crops, to 0.70% when 67 buckets of night soil were used. Infection risk peaked at 0.14% for individuals between the ages of 40–60. Improved sanitation (V) had a net negative, albeit non-linear association with infection.

In 2016–2019, infection probability was stagnant when dry crop area (V) and night soil rice (V) were less than 10 and 23 mu respectively, after which point there was a steep increase in infection probability, peaking at 1.7% at 14 Mus of dry crops (V), and 0.43% at 25 mu of rice (Fig 3). We saw evidence that infection risk was higher in individuals over 80 years of age, and for individuals in households planting >10 Mus of dry crops. Larger areas of rice crops at the village and household level were also associated with increased infection risk.

Table 3 presents the strongest pairwise interactions in 2007–2010 based on our BRT analysis, showing the estimated combined effects of predictors on human schistosomiasis infection risk as measured by their interaction size. The interaction between dry crop area (V) and improved sanitation (V) was the most important interaction, followed by night soil rice crops (V) and dry crop area (H), and bovines (V) and well water (V). The interaction between dry crop area (V) and improved sanitation (V) had a moderate negative relationship (Fig 4). Infection risk was highest for individuals living in households that simultaneously reported planting moderately large areas of dry crops (15 – 17 mu) and were surrounded by households reporting high night soil usage on rice crops (≥60 buckets). Infection risk was also higher for individuals

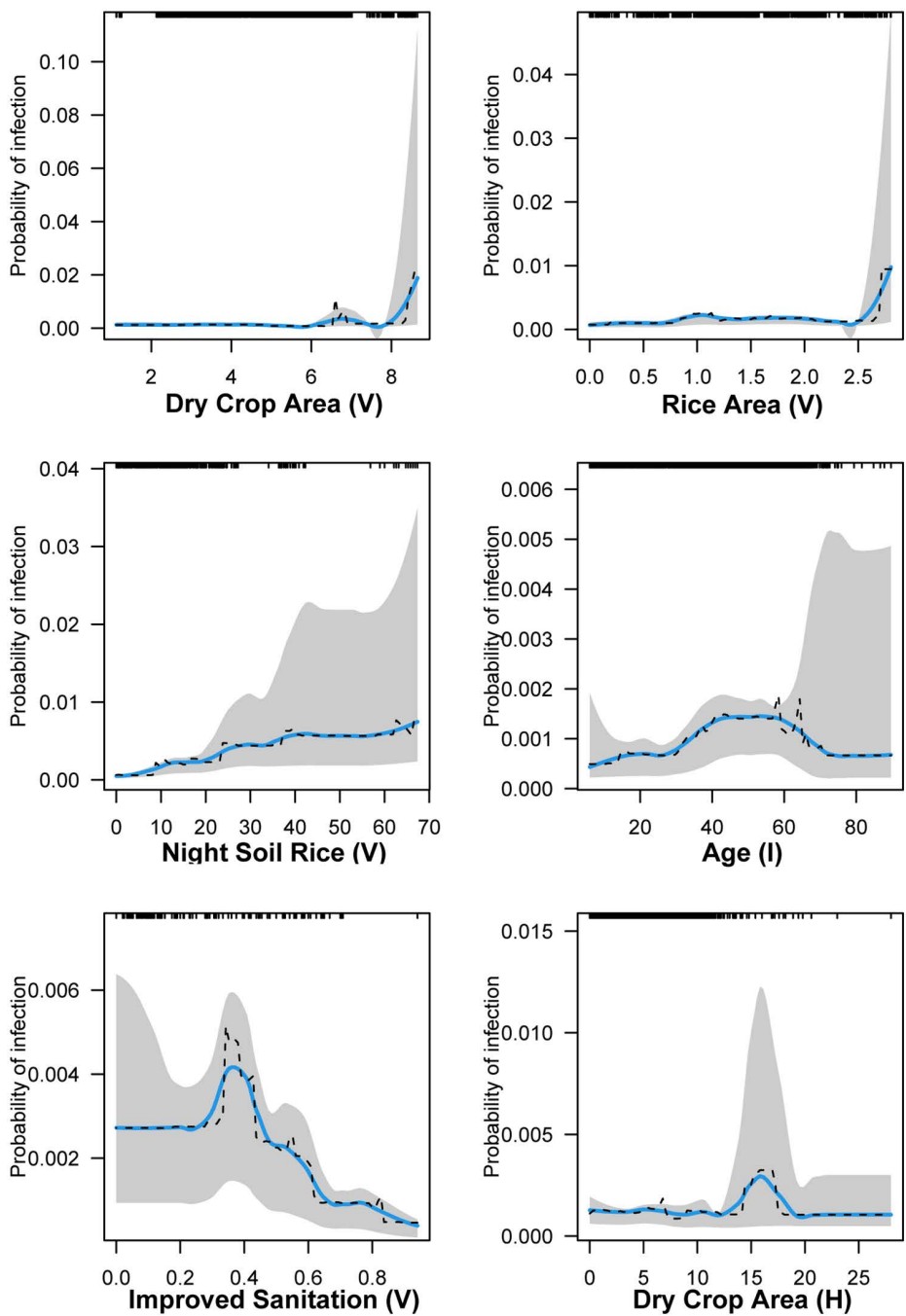

**Fig 2. Partial dependence plots (PDP) of the six most important predictors of human schistosomiasis infection risk in 2007-2010 (reemergence model).** The PDPs display the change in the average predicted infection risk as predictors vary over their marginal distribution while holding all other variables constant. Fitted curves (dashed lines), smoothing splines (solid blue lines) and 95% confidence intervals based on 1000 bootstrap replicates of the data set (shading) are shown. The full distribution of the predictors is displayed as rug ticks on the top of the plot.

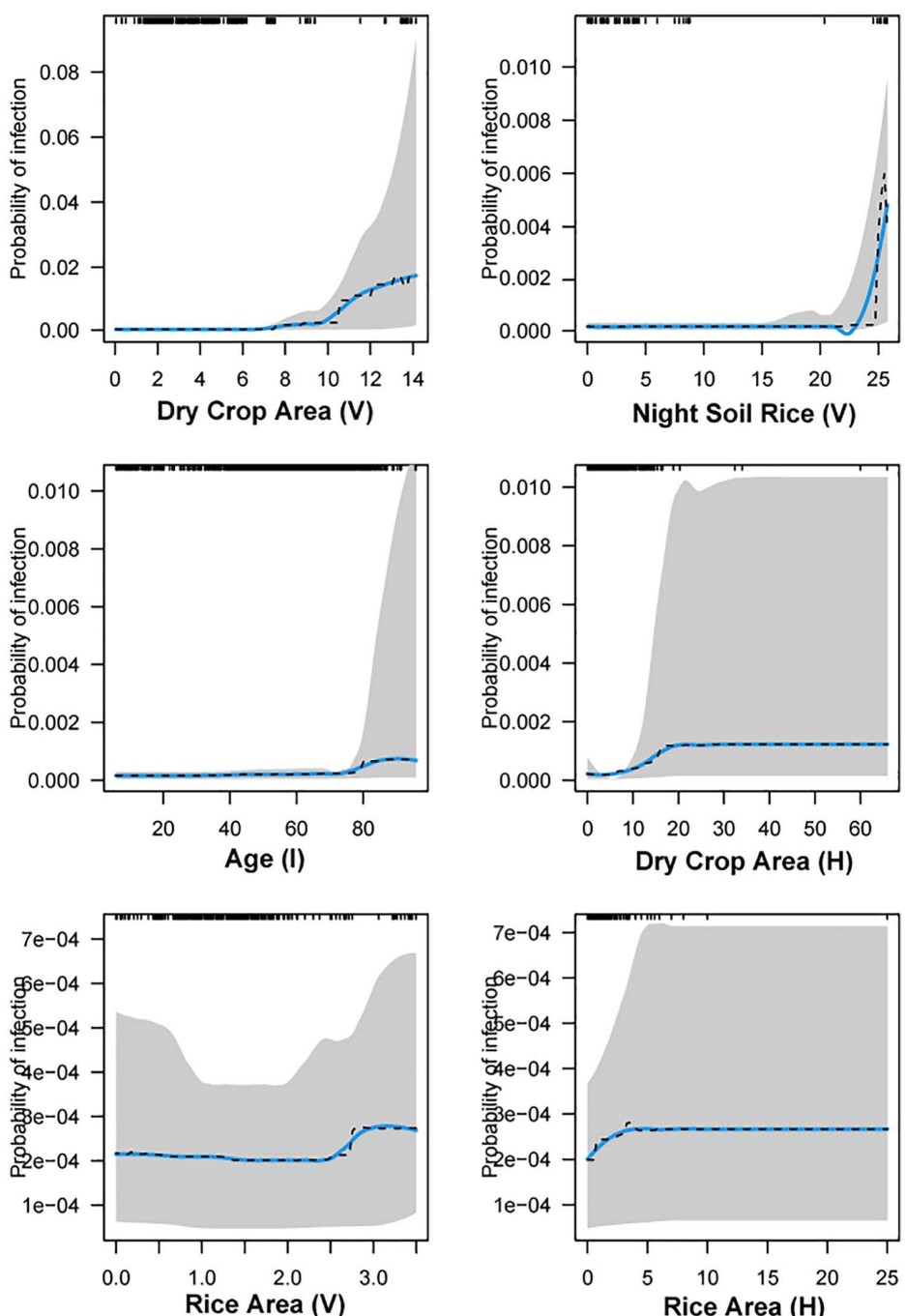

**Fig 3. Partial dependence plots (PDP) of the top six predictors of human schistosomiasis infection risk in 2016-2019 (elimination model).**
The PDP (blue line) displays the change in the average predicted infection risk as predictors vary over their marginal distribution while holding all other variables constant. Fitted curves (dashed lines), smoothing splines (solid blue lines) and 95% confidence intervals based on 1000 bootstrap replicates of the data set (shading) are shown. The full distribution of the predictors is displayed as rug ticks on the top of the plot.

**Table 3. Pairwise interactions that were the strongest predictors of S. japonicum infection (interaction size > 5) in the reemergence period, 2007-2010.**

| Pairwise Interactions | | Interaction Size |
|---|---|---|
| Dry Crop Area (V) | Improved Sanitation (V) | 14.01 |
| Night Soil Rice Crops (V) | Dry Crop Area (H) | 12.96 |
| Bovines (V) | Well Water (V) | 12.81 |
| Rice Crops (V) | County | 8.48 |
| Cats (V) | Bovines (H) | 7.30 |
| Improved Sanitation (V) | County | 6.07 |
| Bovines (V) | Well Water (H) | 5.76 |
| Rice Crops (V) | Bovines (V) | 5.30 |
| Improved Sanitation | Dry Crop Area (H) | 5.06 |

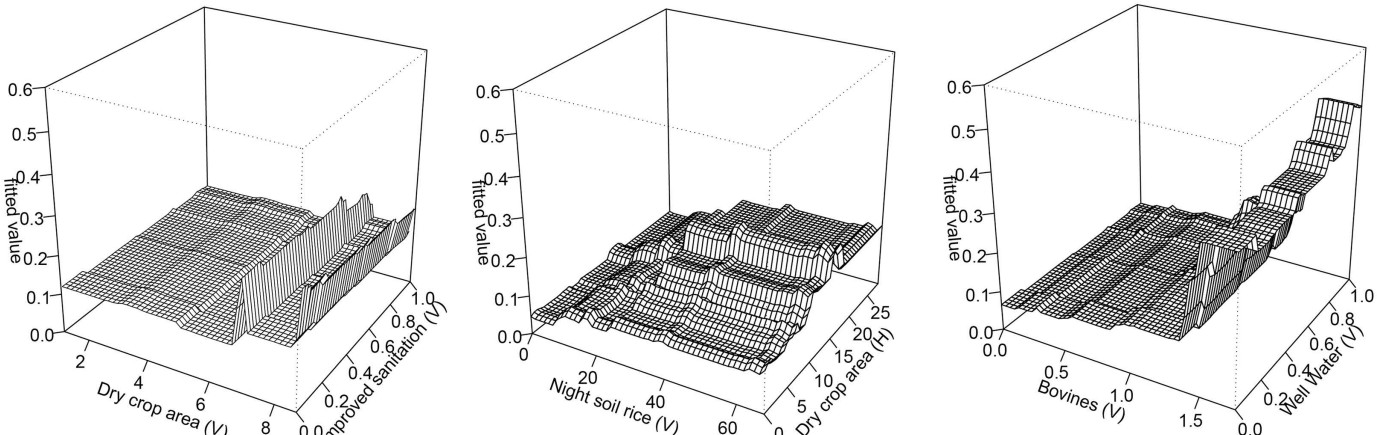

**Fig 4. Three-dimensional partial dependence plots of the three most important pairwise interactions in 2007-2010.** Each panel shows the GBM's fitted value (z-axis; higher values indicate higher predicted *S. japonicum* infection risk) as a function of two predictors (x- and y-axes) with all other covariates averaged over their observed distributions. Left-to-right: (A) village-level dry crop area and village-level improved sanitation; (B) village-level night soil use to rice fields and dry crop area; (C) village-level bovine presence and village-level well-water use.

living in villages with a higher presence of bovines (average household ownership >1.5) and higher well water usage (100%).

Table 4 presents the highest ranked pairwise interactions in 2016–2019 based on our BRT analysis, showing the combined effects of various predictors on human schistosomiasis infection risk, measured by their interaction size. The interaction between the dry crop area (V) and improved sanitation (H) was the most important interaction, followed by dogs (H) and age (I), then dry crop area (V) and age (I). Three of the six most important pairwise interactions included dry crop area (V) and age (I). Infection risk was highest for those over 80 who also owned dogs, or those over 80 who also lived in villages where the reported area of dry crop land was high (>10 mu) (Fig 5).

## Discussion

Our findings suggest that the environmental and socioeconomic predictors of *S. japonicum* transmission are dynamic, with predictor importance shifting as the disease moves from reemergence to near elimination, but there was consistency in the predictive contribution of agricultural land use. Dry crop area (V), rice area (V), night soil rice (V) consistently emerged

**Table 4. Pairwise interactions that were the strongest predictors of S. japonicum infection (interaction size > 5) in the elimination period, 2016-2019.**

| Pairwise Interactions | | Interaction Size |
|---|---|---|
| Dry Crop Area (V) | Improved Sanitation (H) | 33.36 |
| Dogs (H) | Age (I) | 19.96 |
| Dry Crop Area (V) | Age (I) | 18.46 |
| Night Soil Dry Crops (V) | Dogs (H) | 12.54 |
| Night Soil Rice Crops (V) | Dry Crop Area (V) | 12.05 |
| Cats (V) | Age (I) | 11.98 |
| Dry Crop Area (H) | Cats (H) | 9.05 |
| Well Water (V) | Cats (H) | 6.27 |

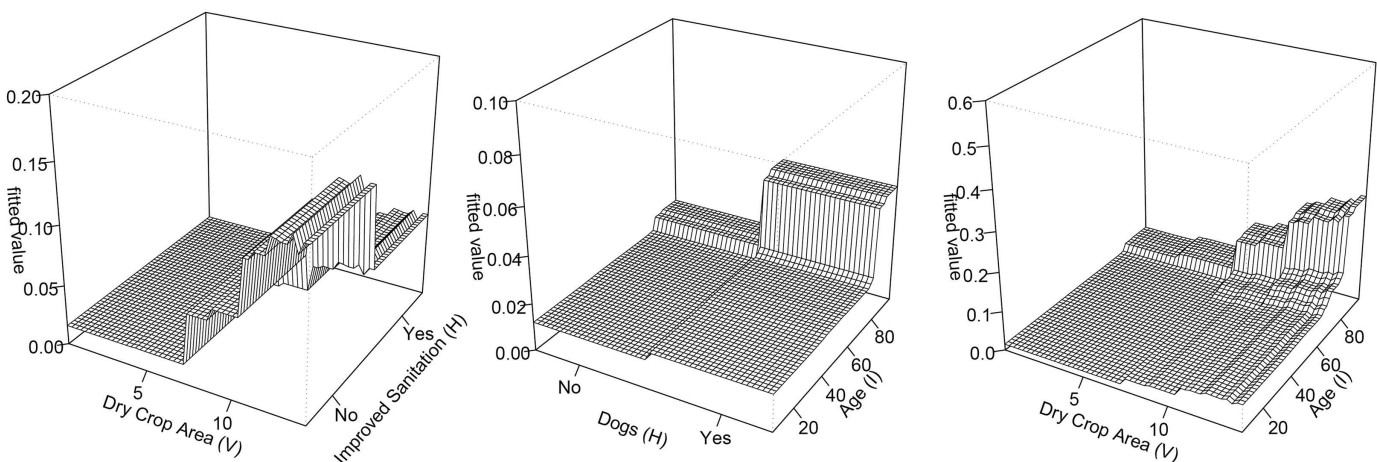

**Fig 5. Three-dimensional partial dependence plots of the three most important pairwise interactions in 2016-2019.** Each panel shows the GBM's fitted value (z-axis; higher values indicate higher predicted *S. japonicum* infection risk) as a function of two predictors (x- and y-axes), with all other covariates averaged over their observed distributions. Left-to-right: (A) village-level dry crop area and household improved sanitation; (B) household dog presence and individual age; (C) village-level dry crop area and individual age.

as a strong predictor across both time periods, indicating that farming practices likely maintain a role in schistosome exposure. Dry crops such as maize, wheat, and rapeseed are often grown on raised embankments or in rotation with paddy rice, creating seasonally moist pockets that may allow snail hosts to persist even when surrounding fields appear dry and may also support other potential reservoirs [27]. Rice paddies, by contrast, require farmers and livestock to spend prolonged periods in water, increasing exposure opportunities [28]. Night soil use may contribute to infection risk by depositing viable *S. japonicum* eggs directly into snail habitats while providing soil and water with organic matter that further boosts snail survival [28]. Night soil use emerged as one of the most influential predictors in both the re-emergence and elimination phases of our analysis. Its persistent importance aligns with other studies showing associations between night soil use and human schistosomiasis infections [29].

While broader indicators of village-level agriculture (e.g., village-level dry crop area) were stronger predictors during the reemergence period, individual and household-level variables (e.g., household-level dry crop area and rice crop area) rose in importance during the elimination period. This trend may reflect a transition from widespread environmental exposure to more specific, household-based transmission risks, suggesting that granular control strategies may be needed

as the *S. japonicum* approaches elimination. The importance of variables describing sanitation also shifted over time. In the reemergence period, village-level improved sanitation (e.g., the proportion of households in the village with improved toilets) was moderately predictive of infection risk, suggesting that broad community-level improved sanitation access shaped transmission dynamics during this time. However, the importance of village-level sanitation was markedly lower in the elimination period when far more participants lived in households with improved sanitation (19% in 2007 vs 55% in 2019). In contrast, household-level sanitation (access to a biogas or three-compartment toilet) became a more important predictor during the elimination phase. This shift may reflect a narrowing of transmission pathways: as widespread environmental contamination declines due to village-level improvements, remaining infections may be increasingly influenced by practices and exposure within a given household which may be directly related to sanitation or sanitation may be a proxy for other variables such as poverty and/or water contact. It may also suggest that community-level sanitation interventions reach a point of saturation and limit their predictive value when high levels of coverage are attained.

The unexpected rise in importance of cat and dog ownership during the elimination period also requires further exploration. One hypothesis is that cats and dogs play distinct roles in household-level transmission ecology. For example, cat ownership might be protective if they control rodent populations that serve as *S. japonicum* reservoir hosts. Recent research indicates that rodents may have become important zoonotic reservoirs of *S. japonicum* in endemic regions of China, with increased infection prevalence in mountainous regions such as Sichuan [9]. Conversely, dogs may be associated with higher risk due to their limited role in rodent control and their own susceptibility to *S. japonicum* infection [10]. Dogs are more likely than cats to accompany people outdoors, enter irrigation channels or streams, and spend time in or near water, which could increase their exposure to cercariae in snail-infested habitats and, by extension, their potential to contribute to transmission. It would be worthwhile to evaluate the consistency of these findings in other contexts. Concurrent testing of humans and other animal hosts in this context would be ideal for future studies.

Variables that were most important for predicting schistosomiasis infection at the individual-level were age in the reemergence and elimination period. Age generally had a positive relationship with infection risk in both time periods, but peak risk shifted rightward to older individuals over time, from a peak risk that occurred at 40–60 years of age in the reemergence period to a peak risk that occurred beyond 80 years of age in the elimination period. One possible explanation for this is the rapid urbanization of the country that may have led to younger individuals working in urban areas while older individuals stayed in rural villages to continue farming. The age distribution between the two periods supports this hypothesis, with a more right-skewed distribution in the earlier years and a more left-skewed distribution in the later period (rug ticks, Figs 2,3; S1) showing aging village populations. Recent literature on the sustainability of farms in China provides further evidence for this finding, with younger farmers realizing "higher incomes by working in non-agricultural sectors in cities" [30]. Our prior work found that travel was associated with lower water contact [17], suggesting that the older populations who are left in the villages are bearing the highest water contact levels. The elimination period also saw education become a more important predictor of schistosomiasis infection, with our results suggesting that those who experienced lower to no levels of schooling had a slightly higher infection risk. The mechanism behind this may be that individuals who received less education may be more likely to work in the agricultural sector and thus risk exposure to *S. japonicum*.

Although our models achieved good predictive performance, several limitations merit consideration. First, class imbalance, particularly the small number of infected cases in 2019 (n = 7), could bias estimates and limit generalizability. As such, our findings from 2019 should be interpreted as hypothesis-generating. We addressed imbalance using Random Walk Oversampling, but synthetic examples may not fully capture true heterogeneity in infection risk. Second, while we used cross-validation and bootstrapping to prevent overfitting, the ensemble nature of boosted models means that overfitting cannot be entirely ruled out. Third, missingness in key predictors may have introduced bias despite imputation, particularly in variables derived from self-reported surveys. Fourth, many variables were drawn from household surveys that were administered with changing survey instruments (e.g., different survey questions regarding seasonal crops) over

the past decade, although we tried to ensure variable calculations were comparable over the study period. Fifth, the comparability of infection estimates across years may be influenced by differences in survey timing (summer in 2016, fall in all other years) and small changes in diagnostic protocols (in 2007 and 2010 we screened all participants with Kato-Katz and miracidial hatch testing, and later limited Kato-Katz testing to those with at least one positive hatch test because of the high labor and low added value of the test in our population). Because schistosomiasis transmission risk and praziquantel treatment schedules can vary seasonally, differences in survey timing could have influenced measured infection outcomes and, in turn, model predictions for 2016. We did not have sufficient repeated within-year infection measurements to formally adjust for seasonality.

Sixth, the generalizability of our results to other endemic regions is uncertain; the agroecological and control context of our study area may differ substantially from other settings in China or Southeast Asia and should be considered in future studies. Our goal was to provide an analytical model that could be replicated in other contexts. Further validation with external datasets is necessary to assess the generalizability of our findings to other areas in China and beyond. Seventh, this study does not provide province-representative prevalence estimates for Sichuan, as it was designed to explore characteristics of persistence in higher-risk settings, selecting and adding villages to the sample where transmission was suspected despite local control efforts.

A key strength of using GBM is its ability to model complex, non-linear relationships and interactions across multiscale predictors, while simultaneously adjusting for a wide set of measured covariates that may confound observed associations. In this sense, the analysis may serve as a useful step toward causal understanding and, ultimately, the design of targeted interventions by identifying predictors and conditions that are consistently associated with infection. However, GBM results should be interpreted with caution: variable importance and response patterns reflect predictive contribution within the observed data structure rather than causal effects. As with any observational analysis, residual confounding from unmeasured or poorly measured factors, measurement error, and potential selection or surveillance biases may influence inferred predictor rankings. Consequently, the findings are best viewed as evidence to prioritize mechanisms and interventions for more targeted causal studies rather than definitive estimates of causal impact.

Overall, our findings support the need for adaptive, context-specific control strategies as schistosomiasis moves toward elimination. Agricultural exposures remain central but increasingly heterogeneous, while the importance of sanitation and companion animal presence appears to evolve over time. Beyond China, this modeling framework is directly applicable to other endemic regions, including Brazil and sub-Saharan Africa, where programs must increasingly prioritize targeted interventions as prevalence declines. By integrating routinely collected infection data with environmental, WASH, and livelihood indicators, similar analyses could help identify persistent transmission "hot spots," and begin to tailor surveillance and elimination strategies to the most important drivers of infection in local transmission environments. A next step for future analyses would be to translate these predictors into an implementation-ready risk stratification tool and evaluate, from a program perspective, whether risk-based targeting reduces costs and staff burden while maintaining sensitivity compared with population-wide surveillance in low-prevalence settings.

## Conclusion

Our findings provide insights into the shifting epidemiology of *S. japonicum* infection in Sichuan province across two different phases of disease control. Our boosted regression models allowed us to explore the associations and interactions between agricultural, socioeconomic, and individual risk factors at multiple spatial scales. The shift from village-level predictors being dominant during the reemergence period to a more balanced distribution of influential predictors across village, household, and individual-levels during the elimination period suggests that as transmission declines, more localized and individualized factors play a greater role in determining infection risk. This aligns with broader trends of urbanization

and shifting agricultural practices in China, where younger individuals are leaving rural areas, leading to an older and potentially more exposed farming population.

These findings are intended to inform surveillance and control prioritization rather than to estimate causal effects. In an elimination setting, the persistent importance of agriculture-related indicators, particularly night soil and paddy-associated measures, supports focusing case detection and environmental monitoring in villages and households where these practices remain. More broadly, our results reinforce the need for adaptive strategies that evolve with the epidemiological phase of control: community-wide interventions may be most impactful during reemergence, while elimination may require supplemental targeted interventions.

## Supporting information

**S1 File. Sampling Strategy.** A comprehensive description of our sampling strategy.
(PDF)

**S1 Table. A summary and description of the predictors included in our analysis.**
(PDF)

**S1 Fig. Age distribution of participants: histograms for each of the four survey years.**
(PNG)

**S2 Fig. Heatmap of pearson correlation coefficients between all predictors retrieved from village-level, household and individual demographic and infection surveys.**
(PNG)

## Acknowledgments

We thank the late Dr. Robert Spear for his mentorship and early work on schistosomiasis in Sichuan that laid the foundation for this study.

## Author contributions

**Conceptualization:** William W. Zou, Liu Yang, Elizabeth J. Carlton.

**Data curation:** Elise N. Grover, Liu Yang, Elizabeth J. Carlton.

**Formal analysis:** William W. Zou.

**Funding acquisition:** Liu Yang, Elizabeth J. Carlton.

**Investigation:** William W. Zou.

**Methodology:** William W. Zou, Elise N. Grover.

**Project administration:** Liu Yang, Elizabeth J. Carlton.

**Resources:** Liu Yang, Elizabeth J. Carlton.

**Software:** William W. Zou, Elise N. Grover.

**Supervision:** Elise N. Grover, Liu Yang, Elizabeth J. Carlton.

**Validation:** William W. Zou.

**Visualization:** William W. Zou.

**Writing – original draft:** William W. Zou, Elizabeth J. Carlton.

**Writing – review & editing:** William W. Zou, Elise N. Grover, Liu Yang, Elizabeth J. Carlton.

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
