## [Decision Letter · Decision Letter 0]

11 Nov 2025

Response to Reviewers

* A marked-up copy of your manuscript that highlights changes made to the original version. You should upload this as a separate file labeled 'Revised Manuscript with Track Changes '.

* An unmarked version of your revised paper without tracked changes. You should upload this as a separate file labeled 'Manuscript '.

We look forward to receiving your revised manuscript.

Kind regards,

Robert Adamu SHEY, Ph.D.

Guest Editor

Jong-Yil Chai

Section Editor

Shaden Kamhawi

co-Editor-in-Chief

Paul Brindley

co-Editor-in-Chief

Dear Editor,

Please kindly respond to the comments of the peer reviewers as you revise the manuscript. In addition, it is important to include in the introduction the current global burden of schistosomiasis and its potential impact before focusing on China. In addition, make sure that all scientific names are written following the systematics of nomenclature.

Kind regards

**Journal Requirements:**

At this stage, the following Authors/Authors require contributions: William Wei Zou, Elizabeth J Carlton, Elise N Grover, and Liu Yang. Please ensure that the full contributions of each author are acknowledged in the "Add/Edit/Remove Authors" section of our submission form.

Potential Copyright Issues:

i) Please confirm (a) that you are the photographer of Graphical Abstract, or (b) provide written permission from the photographer to publish the photo(s) under our CC BY 4.0 license.

5) We note that you have indicated that there are restrictions to data sharing for this study. PLOS only allows data to be available upon request if there are legal or ethical restrictions on sharing data publicly. For more information on unacceptable data access restrictions, please see https://journals.plos.org/plosntds/s/data-availability#loc-unacceptable-data-access-restrictions.

b) If there are no restrictions, please upload the minimal anonymized data set necessary to replicate your study findings to a stable, public repository and provide us with the relevant URLs, DOIs, or accession numbers. For a list of recommended repositories, please see https://journals.plos.org/plosone/s/recommended-repositories. You also have the option of uploading the data as Supporting Information files, but we would recommend depositing data directly to a data repository if possible.

**Reviewers' comments:**

Reviewer's Responses to Questions

**Key Review Criteria Required for Acceptance?**

**Methods**

-Are the objectives of the study clearly articulated with a clear testable hypothesis stated?

-Is the study design appropriate to address the stated objectives?

-Is the population clearly described and appropriate for the hypothesis being tested?

-Is the sample size sufficient to ensure adequate power to address the hypothesis being tested?

-Were correct statistical analysis used to support conclusions?

-Are there concerns about ethical or regulatory requirements being met?

Reviewer #1: (No Response)

Reviewer #2: The study’s objectives are clearly articulated and aligned with its elimination-focused context, though the hypotheses could be more explicitly framed to reflect correlation-based rather than causal inference. The overall design, integrating multiscale (individual, household, and village) data, is appropriate and methodologically strong. The study population is well described and relevant to the research question.

The statistical approach (GBM) is suitable for exploring complex relationships, additional details on model hyperparameters, variable selection, and sensitivity analyses would improve transparency and rigor. The sample size appears adequate, though more discussion on representativeness and site-specific limitations would be helpdul. There are no apparent ethical or regulatory concerns.

Reviewer #3: The methods are well written and no revisions are necessary

**Results**

-Does the analysis presented match the analysis plan?

-Are the results clearly and completely presented?

-Are the figures (Tables, Images) of sufficient quality for clarity?

Reviewer #1: (No Response)

Reviewer #2: The analyses presented overall align with the study’s objectives and design. Results are clearly organized and correspond to the stated analytical framework, though additional transparency regarding GBM model parameters and sensitivity analyses would strengthen confidence in the findings. Figures and tables are of acceptable quality but could be improved by ensuring captions are fully interpretive and that performance metrics include accompanying p-values or confidence intervals for clarity

Reviewer #3: The results are well presented and no revisions are necessary

**Conclusions**

-Are the conclusions supported by the data presented?

-Are the limitations of analysis clearly described?

-Do the authors discuss how these data can be helpful to advance our understanding of the topic under study?

-Is public health relevance addressed?

Reviewer #1: (No Response)

Reviewer #2: The conclusions are overall supported by the data presented, although the discussion at times 'overinterprets' correlations as causal relationships. Reframing this or re-wording this a bit would improve the rigor of the paper. The limitations of the analysis, particularly the site-specific sampling and model constraints, should be discussed more explicitly. Strengthening the discussion on how findings inform surveillance and control strategies would enhance its public health relevance and broader impact.

Reviewer #3: Minor edits are suggested (see general comments)

**Editorial and Data Presentation Modifications?**

Reviewer #1: (No Response)

Reviewer #2: Acceptable

Reviewer #3: (No Response)

**Summary and General Comments**

Reviewer #1: This research, titled "One Health at the Last Mile: Multi-scale Predictors of Schistosoma japonicum Infection in Southwest China across Two Decades of Control," employs a Gradient Boosting Machine (GBM) model to analyze the risk factors for schistosomiasis infection in Sichuan Province, China, during two critical periods: the re-emergence phase (2007–2010) and the near-elimination phase (2016–2019). By integrating multi-scale variables at the individual, household, and village levels—such as agricultural practices, animal hosts, socioeconomic status, and water sanitation facilities—the study reveals significant shifts in the relative importance of risk factors as control efforts progressed. It finds that in the near-elimination phase, household-level factors like agricultural activities and pet ownership became more influential, while the impact of village-level water sanitation facilities diminished. The study emphasizes the need for targeted intervention strategies during the elimination phase, focusing on high-risk households and agricultural practices.

Innovations:

Multi-scale analysis: The study combines variables at the individual, household, and village levels to comprehensively assess the risk of schistosomiasis transmission.

Temporal comparison: By comparing the re-emergence and near-elimination phases, it uncovers dynamic evolutionary patterns in risk factors.

One Health perspective: It integrates human, animal, and environmental factors, aligning with the "One Health" concept.

Policy implications: Provides empirical evidence for precision interventions and monitoring strategies during the elimination phase.

Some issues in the study require reconsideration:

Model generalization: The data were sourced from specific high-risk villages in Sichuan Province, and the number of infection cases in 2019 was extremely low (n=7). Did the authors consider validating the model’s generalizability through cross-validation or external datasets? How did they ensure the conclusions are applicable to other low-endemic areas?

Variable selection and collinearity: The study includes multiple agricultural and environmental variables, which may be highly correlated (e.g., crop area and fertilization methods). How did the authors address collinearity among variables? Did they assess the impact of collinearity on variable importance rankings?

Seasonal bias: The infection survey in 2016 was conducted in summer, while other years were surveyed in winter. Could seasonal differences have affected infection detection results and model predictions? Did the authors attempt to adjust for or address this potential bias?

Mechanism of animal hosts: The study found that livestock rearing became more important in the elimination phase, but the mechanism remains unclear. The authors suggest that cats may reduce transmission by controlling rodents, while dogs may increase risk. Is there further data or analysis to support these hypotheses? Were animal infection detection data considered?

Age-related risk shift: The high-risk age group shifted from 40–60 years to 80 years and older, which the authors attribute to the outflow of young rural populations. Is there direct data on population mobility or occupational exposure to support this explanation? Were behavioral pattern changes among the elderly considered?

Feasibility of intervention strategies: The authors recommend increasing targeted interventions for high-risk households during the elimination phase. How can these households be identified in practice? Is it feasible to implement such precision measures in resource-limited areas? Was a cost-effectiveness analysis conducted?

Data missingness and imputation: The study had 3.27% missing data, which was imputed using random forests. Was the impact of the imputation method on the stability of the results evaluated? Was a sensitivity analysis conducted to address potential biases introduced by missing data?

Reviewer #2: Overall a strong manuscript. Some issues, minor concerns are described in the section above already.

Reviewer #3: This is a well written, interesting and informative paper looking at the predictions of S. japonicum transmission at different stages of control and elimination. The analyses seems robust and the outcomes will be very useful for interventions going forward.

The introduction would benefit from more background to include

• The main human species – Africa and Asia and the global burden (not just China)

• The key feature of transmission in China (zoonotic transmission)

• More information on the different animal hosts and transmission environments

• The snails, environment and general life-cycle (how transmission happens)

• Clearly define S. japonicum as having bovines as reservoirs but that S. japonicum is zoonotic – although they are connected they have different definitions.

• Define elimination (e.g. as a public health problem or transmission interruption)

Line 64 – The terminology should be intestinal schistosomiasis caused by S. japonicum.

Line 123 – add detail as to what the animal reservoirs were

Discussion

• It would be good to also mention the different behaviours of cats and dogs in terms of their involvement S. japonicum transmission. e.g. dogs like water and so are more likely to be exposed compared to cats.

• It would be good to have a sentence or two on the boarder aspects of the work beyond Africa. How could this type of work support Schistosomiasis control and elimination on Brazil and Africa.

General – in various places the words “parasite” or “schistosome” are used whereas the authors should be more specific and use S. japonicum. Also, check that all species names are in Italics.

PLOS authors have the option to publish the peer review history of their article (what does this mean? ). If published, this will include your full peer review and any attached files.

**Do you want your identity to be public for this peer review?** For information about this choice, including consent withdrawal, please see our Privacy Policy .

Reviewer #1: No

Reviewer #2: No

Reviewer #3: No

**Figure resubmission:**
---

## [Editor Report · Decision Letter 1]

26 Jan 2026

Dear Dr. Carlton,

We are pleased to inform you that your manuscript 'One Health at the Last Mile: Multi-scale Predictors of Schistosoma japonicum Infection in Southwest China across Two Decades of Control' has been provisionally accepted for publication in PLOS Neglected Tropical Diseases.

Best regards,

Robert Adamu SHEY, Ph.D.

Guest Editor

Jong-Yil Chai

Section Editor

Shaden Kamhawi

co-Editor-in-Chief

Paul Brindley

co-Editor-in-Chief

---

## [Editor Report · Acceptance letter]

Dear Dr. Carlton,

We are delighted to inform you that your manuscript, "One Health at the Last Mile: Multi-scale Predictors of Schistosoma japonicum Infection in Southwest China across Two Decades of Control," has been formally accepted for publication in PLOS Neglected Tropical Diseases.

Best regards,

Shaden Kamhawi

co-Editor-in-Chief

Paul Brindley

co-Editor-in-Chief
